

# Spectral Aerosol Optical Depth from Traceable Spectral Solar Irradiance Measurements to the SI

Julian Gröbner[1], Natalia Kouremeti[1], Gregor Hülsen[1], Ralf Zuber[2], Mario Ribnitzky[2], Saulius Nevas[3],
Peter Sperfeld[3], Kerstin Schwind[3], Philipp Schneider[3], Stelios Kazadzis[1], África Barreto[4],
Tom Gardiner[5], Kavitha Mottungan[5], David Medland[5], and Marc Coleman[5]

[1]Physikalisch-Meteorologisches Observatorium Davos, World Radiation Center (PMOD/WRC), Davos Dorf, Switzerland
[2]Gigahertz Optik GmbH, Türkenfeld by Munich, Germany
[3]Physikalisch-Technische Bundesanstalt (PTB), Braunschweig, Germany
[4]Izaña Atmospheric Research Center (IARC), State Meteorological Agency (AEMET), Santa Cruz de Tenerife, Spain
[5]National Physical Laboratory (NPL), Teddington, UK

**Correspondence:** Julian Gröbner (julian.groebner@pmodwrc.ch)

**Abstract.** Spectroradiometric measurements of direct solar irradiance traceable to the SI were performed by three spectrora-diometer systems during a three week campaign in September 2022 at the Izaña Atmospheric Observatory (IZO) located on the island of Tenerife, Canary Islands, Spain. The spectroradiometers provided direct spectral irradiance measurements in the spectral range 300 nm to 550 nm (QASUME), 550 nm to 1700 nm (QASUME-IR), 300 nm to 2150 nm (BTS), and 316 nm

to 1030 nm (PSR), with relative standard uncertainties of 0.7%, 0.9% and 1% for QASUME/QASUME-IR, PSR, and BTS respectively. The calibration of QASUME and QASUME-IR was validated prior to this campaign at the PTB by measuring the spectral irradiance from two spectral irradiance sources, the high temperature blackbody BB3200pg as national primary stan-dard and the tuneable laser facility TULIP. The Top of Atmosphere (ToA) solar irradiance spectra from the spectroradiometers were retrieved from direct solar irradiance measurements using zero airmass extrapolation during cloudfree conditions which

were then compared to the TSIS-1 HSRS solar spectrum. These ToA solar spectra agreed to within 1% for the spectral range longer than 400 nm (for QASUME also at shorter wavelengths) in the spectral regions free of significant trace gas absorption and well within the combined uncertainties over the full investigated spectral range. Using the results from the comparison with QASUME, the relative standard uncertainty of the TSIS-1 HSRS ToA solar spectrum in the spectral range 308 nm to 400 nm could be reduced from its nominal 1.3% to 0.8%, representing the relative standard uncertainty of the QASUME ToA solar

spectrum in this spectral range. The spectral Aerosol Optical Depth (AOD) retrieved from the solar irradiance measurements of these spectroradiometers using TSIS-1 HSRS as the reference ToA solar spectrum agreed to within 0.01 in optical depth in nearly all common spectral channels of two narrowband filter radiometers belonging to the GAWPFR and AERONET net-works. This study shows that it is now possible to retrieve spectral AOD over the extended spectral range from 300 nm to 1700 nm using solar irradiance measurements traceable to the SI using laboratory calibrated spectroradiometers with similar qual-

ity as from traditional Langley-based calibrated instruments. The main improvement to previous investigations is the recent availability of the high spectral resolution TSIS-1 HSRS solar spectrum with very low uncertainties which provides the Top



of Atmosphere reference for the spectral atmospheric transmission measurements obtained from ground based solar irradiance measurements.

## 1 Introduction

Atmospheric aerosols are minor constituents of the atmosphere, but an important component in terms of impacts on the climate. Their properties have been recognized as Essential Climate Variables (ECVs) by the Global Climate Observing System (GCOS) of the World Meteorological Organisation (WMO). As pointed out in all IPCC reports, aerosols continue to contribute the largest uncertainty to estimates and interpretations of the Earth's changing energy budget (IPCC, 2021). Long-term monitoring of aerosol ECVs including their uncertainties is needed for observing sensitive changes in the Earth climate system. Aerosol
optical properties from ground-based passive remote sensing radiometers have been retrieved consistently for the past 20 years.

Current aerosol optical remote sensing networks rely either on a strict calibration hierarchy, based on bilateral comparisons between network and reference instruments (GAW-PFR (Kazadzis et al., 2018) and AERONET (Holben et al., 1998)), or on in-situ calibrations of network radiometers without any link to traceable standards (SKYNET (Nakajima et al., 2020)). While the latter procedure precludes any form of metrological traceability, the GAW-PFR and AERONET networks have
well established procedures for the traceability of network radiometers to their reference radiometers. These radiometers use narrowband interference filters in specifically chosen spectral regions unaffected by strong atmospheric gas absorptions to retrieve the spectral Aerosol Optical Depth from spectral transmission measurements of the atmosphere. The crucial element in this process is the knowledge of the solar (or lunar) irradiance at the top of the atmosphere (ToA). The established methodology for obtaining the top of the atmosphere spectral irradiances is through in-situ calibrations of the reference radiometers based
on zero airmass extrapolations (also called Langley-plot procedure) at pristine high altitude sites (Shaw, 1983; Toledano et al., 2018) and references therein. Then, assuming stability of the radiometers, these are relocated to their respective calibration sites (for example PMOD/WRC, Davos, Switzerland for GAWPFR, or Observatoire de Haute-Provence, France, Valladolid and Izaña, Spain, in the case of AERONET-Europe) to transfer their ToA irradiance values to the network radiometers.

This approach is currently the most accurate method for the determination of atmospheric transmission, since only relative
measurements of the same instrument are required to obtain the ToA and the surface irradiances. The corresponding measurement uncertainty of the atmospheric transmission can be quantified by the variability of the retrieved ToA solar spectrum on subsequent half-days. As discussed by Toledano et al. (2018), the standard error of the mean of the retrieved ToA irradiances can be significantly less than 0.5%, as also shown later in this study.

However there are several disadvantages to this methodology:

– The Langley procedure to retrieve the ToA irradiances requires stable atmospheric conditions with constant AOD during at least one half day to extrapolate the irradiance measurements to zero airmass. The procedure assumes that the atmosphere varies randomly during the calibration period and a statistical approach is applied to retrieve a representative ToA irradiance from a set of measurements (Toledano et al., 2018). Systematic atmospheric variations (for example in the short ultraviolet due to ozone related photochemical effects), would go unnoticed and potentially falsify the results. Fur-



thermore, there is a trade-off between the length of the calibration period and the observed degradation of the instrument which is difficult to quantify and varies for different instruments.

– The zero airmass extrapolation fails in spectral regions with saturated trace gas absorptions, as for example in the water vapour regions around 930 nm.

– Metrological traceability of the reference radiometers is lost after their relocation from the high altitude site to their respective nominal operation sites. The calibration validity is instead assessed through instrument redundancy, and assumptions about instrument stability and degradation.

– Since the irradiance measurements of the radiometers are not traceable to the SI, the solar irradiance measurements themselves cannot be used for quality control or comparisons between instruments.

Since the 1970's, stratospheric balloon, rocket and then satellite based experiments have been measuring the top-of-atmosphere
spectral solar irradiance with negligible atmospheric absorption. In principle, these ToA solar spectra, given in SI units, typically in $Wm^{-2}nm^{-1}$, therefore provide directly the necessary ToA irradiances needed to retrieve the atmospheric transmission from calibrated ground-based solar irradiance measurements. However uncertainties in the ToA solar spectra and of the ground-based spectral measurements have so far been too large to achieve the desired uncertainties in atmospheric transmission and thus AOD, when compared to the self-consistent approach based on the zero airmass extrapolation for each instrument.

The situation has considerably improved with the advent of fully SI-traceable characterised and calibrated satellite experiments such as TSIS-1 with a combined standard uncertainty of less than 0.25% (Richard et al., 2020). Furthermore, a hybrid high spectral resolution solar spectrum based on TSIS-1 measurements has been constructed to provide ToA solar spectra with standard uncertainties between 0.3% and 1.3% over the spectral range from 202 nm to 2370 nm (Coddington et al., 2021). As discussed in Coddington et al. (2021), this hybrid solar spectrum TSIS-1 HSRS agreed to within 0.5% with the ToA solar spec-
trum QASUMEFTS obtained from zero airmass extrapolations from SI-traceable ground-based solar irradiance measurements with the QASUME spectroradiometer in the spectral range 300 nm to 500 nm (Gröbner et al., 2017). Furthermore, Kouremeti et al. (2022) have shown that AOD retrieved from SI-traceable solar irradiance measurements with a precision filter radiometer at the spectral channels 368 nm, 412 nm, 500 nm, and 862 nm, agreed to within 0.01 when compared with the AOD derived using a Langley-plot based calibration of the same instrument.

In this study, we will discuss SI-traceable spectral solar irradiance measurements performed with solar spectroradiometers at the high altitude observatory in Izaña, Tenerife, during a three-week field campaign in the frame of the project EMPIR 19ENV04 MAPP ("Metrology for aerosol optical properties"). The ToA solar spectra derived from zero airmass extrapolations are compared with the TSIS-1 HSRS solar reference spectrum over the range 300 nm to 2100 nm. Furthermore, the spectral aerosol optical depth retrieved from using the TSIS-1 HSRS ToA solar irradiances are compared with the AOD from collocated
measurements from reference filter radiometers from the GAWPFR and AERONET networks.


## 2 Instruments

The measurements were performed at the Izaña Atmospheric Observatory (IZO) located on the island of Tenerife (Canary Island, Spain, 28.309 N, 16.499 W) from 6 to 22 September 2022. IZO is a high mountain station at an elevation of 2373 m above sea level (a.s.l) above a strong subtropical temperature inversion layer, which acts as a natural barrier for local pollution and low-level clouds. The site is a primary calibration site for instruments performing zero airmass extrapolations due to its stable atmospheric conditions during most of the year. Based on a long-term climatology of the site, the period of the campaign was selected so as to offer the highest probability for clear skies, stable total column ozone values and a minor probability of Saharan dust intrusions.

The spectroradiometers deployed at this campaign and used in this study are described in the following sub-sections.

### 2.1 QASUME spectroradiometer

The transportable reference spectroradiometer QASUME is essentially the same instrument as used in the previous campaign in 2016 to retrieve the ToA solar spectrum between 300 nm and 500 nm (Gröbner et al., 2017). It consists of a scanning double monochromator with a focal length of 150 mm and two 2400 lines/mm gratings resulting in a full width at half maximum (FWHM) of 0.86 nm. The whole system resides in a temperature controlled enclosure to allow outdoor operation under varying ambient conditions. The solar radiation is collected with a temperature stabilised diffuser connected via an optical fiber to the entrance slit of the monochromator. A portable lamp monitoring system allows for the calibration of the whole system while being deployed in the field. A detailed description of the system can be found in Gröbner et al. (2005); Hülsen et al. (2016). A collimator tube with a full opening angle of 2.5 ° is mounted on an optical tracker to which the diffuser head can be fitted, allowing the measurement of direct solar spectral irradiance.

One significant improvement to the instrument was the addition of a monitor diode at the output of the monochromator to correct for daily hysteresis changes of up to 1% coming from the photomultiplier. This slightly reduced the uncertainty budget described in (Gröbner et al., 2017) from 1.8% to 1.4% (expanded relative uncertainty, k=2, representing a coverage probability of 95% assuming a normal probability distribution). QASUME was calibrated every day using a portable lamp monitoring system with a set of three 250 W tungsten-halogen lamps in order to verify its stability and confirm its traceability to the SI. The calibrations were performed daily, and varied by less than ±0.5% over the course of the campaign.

### 2.2 QASUME-IR spectroradiometer

The QASUME-IR spectroradiometer consists of a single monochromator with focal length 300 mm. Two gratings are used to cover the extended spectral range from 550 nm to 900 nm (1200 lines/mm), and 900 nm to 1700 nm (830 lines/mm). The corresponding spectral resolution for these two regions is 1.6 nm and 2.4 nm (FWHM). The entrance optic consists of an integrating sphere, coupled to the entrance port of the spectroradiometer with an optical quartz fiber. The integrating sphere is connected to a collimator with the same dimensions as used for QASUME to allow measurements of the direct solar irradiance. Two temperature stabilised photodiodes (Silicon-based until 900 nm and InGaAs for the second spectral range) measure the





dispersed radiation at the two output slits of the monochromator. The instrument was calibrated daily using the same portable
system as used for QASUME, resulting in a similar relative expanded uncertainty of 1.4% (k=2) apart from the spectral bands
120  strongly affected by atmospheric humidity as discussed below.

The responsivity of QASUME-IR was very stable throughout the campaign, with a variability of less than 0.4% in the spec-
tral range 550 nm to 1300 nm. Between 1300 nm and 1500 nm, the measured responsivity varied by 1.5%, due to ambient
changes in relative humidity that absorbed some of the radiation emitted by the transfer standard lamp in the portable moni-
toring system, before reaching the entrance optic of the spectroradiometer. This effect was not corrected since the atmospheric
125  water vapour absorbs nearly all of the solar irradiance in this spectral interval and thus this part of the solar spectrum was not
further used in the analysis.

### 2.3 Precision Solar Spectroradiometer PSR

The Precision Solar Spectroradiometer (PSR) was designed and built at PMOD/WRC (Gröbner and Kouremeti, 2019). It is
based on a temperature stabilized grating spectroradiometer with a 1024 pixel Hamamatsu diode-array detector, operated in
130  a temperature controlled hermetically sealed nitrogen flushed enclosure. The spectroradiometer measures the solar spectrum
in the 316 nm to 1030 nm spectral range with a FWHM of between 1.5 nm and 4 nm. The relative expanded uncertainty
(k=2) is 1.8% for direct normal solar spectral irradiance measurements in the central wavelength range between 400 nm and
900 nm, with slightly increasing uncertainties at either end of the spectrum. At this campaign, PSR#009 was deployed, using
a responsivity calibration performed in the optical laboratory of PMOD/WRC prior and after the campaign using transfer
standards traceable to the primary spectral irradiance standard of the PTB (Gröbner and Sperfeld, 2005). The differences in the
two calibrations were less than 1 % over the whole spectral range, confirming the uncertainty budget described in Gröbner and
Kouremeti (2019).

### 2.4 BTS spectroradiometers

The BTS is a commercially available system composed of two array-spectroradiometers made by the company Gigahertz Optik
GmbH. The spectral range from 300 nm to 1050 nm is covered with a 2048 pixel Si BTS2048-VL-TEC-WP with a nominal
spectral resolution (FWHM) of 2.5 nm whose characterisation was described in Zuber et al. (2018), while the spectral range
from 950 nm to 2150 nm is measured with a BTS2048-IR-WP with a nominal spectral resolution of 8 nm (FWHM) wiht 512
pixel and an extended InGaAs detector. The spectral region of AOD could be extended above 1700 nm to 2150 nm compared
to the other used instruments. Both instruments are temperature stabilised and directly mounted on a solar tracker without any
optical fiber. Each spectroradiometer has a collimator to measure only direct solar irradiance with a diffusor entrance optic.
The BTS2048-IR-WP has been calibrated by the PTB a half year before the campaign with an estimated relative expanded
measurement uncertainty of 3.3% (k=2) up to 1600 nm and 4.3% (k=2) above 1600 nm.

The BTS2048-VL-TEC-WP was calibrated in the ISO 17025 calibration laboratory of Gigahertz-Optik, which is traceable
to PTB and has been validated by the PTB who confirmed the calibration uncertainty. An estimated expanded measurement
uncertainty (k=2) of 3.5% from 300 nm to 330 nm, 1.9% (k=2) from 330 nm to 450 nm and 1.8% (k=2) for the remaining



spectral range was achieved. The calibrations were checked on-site using a commercially available mobile transfer standard based on a 250 W tungsten lamp called BN-LHSI-WP before and after the measurement campaign. The BTS2048-VL-TEC-WP calibration did not change significantly and therefore the laboratory calibration was used during the whole campaign.

## 2.5  Narrowband filter sunphotometers

The AOD from a Precision Filter Radiometer (Wehrli, 2000) and a CIMEL CE318 sunphotometer belonging to the GAWPFR and AERONET networks respectively were used for the comparison and validation of the retrieved spectral aerosol optical depth from the spectroradiometers. The operation, data production and measurement uncertainties are as described in the extensive literature of their networks (Kazadzis et al., 2018; Holben et al., 1998). Noteworthy for their use in this study is that both radiometers are so called reference or master instruments of their respective networks, since Izaña is the calibration site at

which the zero airmass extrapolation is performed. Therefore the AOD obtained from these radiometers is based directly and without any further calibration steps on the zero airmass extrapolation performed at the site without instrument transportation or other activities which could break the traceability.

### 2.5.1  Precision Filter Radiometer PFR N01

A second precision filter radiometer, PFR N01, was also operated during the measurement campaign. The four spectral channels

of this radiometer were fully characterised and calibrated with respect to the SI at the tuneable laser facility of PTB in January 2021 (Kouremeti et al., 2022). As shown in that study, the Top of Atmosphere solar irradiances retrieved with this radiometer agreed very well with TSIS-1 HSRS. A slight decrease in responsivity of PFR N01 was observed between its calibration at PTB in January 2021, linked with the comparison to the PFR Triad in September 2020 and its deployment during this campaign in September 2022. This degradation was monitored by intermediate comparisons to the PFR Triad at PMOD/WRC (Kazadzis

et al., 2018) and verified by a Langley calibration. These comparisons showed that the responsivity of PFR N01 decreased in these 2 years by 0.12%, 0.35%, 0.52%, and 0.62% for the spectral channels 862 nm, 500 nm, 411 nm, and 368 nm respectively. This change in responsivity was taken into account when calculating the solar irradiances of PFR N01 during this campaign.

## 2.6  Solar-pointing Fourier Transform Infrared Spectrometer

A solar-pointing Fourier Transform Infrared Spectrometer (FTIR) was also deployed at IZO during the campaign. This in-

strument was a scanning Michelson design supplied by Bruker Corp. (model 125M) coupled to a solar tracking system (also supplied by Bruker). It was used to provide high-resolution (0.01 cm$^{-1}$) measurements of the solar spectrum between 500 nm and 5500 nm. Langley analysis based on Kiedron and Michalsky (2016) was used to separate the top-of-atmosphere and atmospheric contributions to the measurements across the near-infrared region between 950 nm to 2100 nm. This instrument has previously been used in the evaluation of the near-infrared top-of-atmosphere spectrum (Elsey et al., 2017) and assessment

of the water vapour continuum in the near-infrared region (Elsey et al, 2020).



# 3 Spectral irradiance traceability to the SI

The spectral irradiance measurements of the spectroradiometers used in this study are ultimately traceable to the high temperature blackbody at PTB, acting as German national standard for spectral irradiance. The spectral irradiance measurements at PMOD/WRC are traceable to that primary spectral irradiance standard via a set of transfer standard lamps, with an estimated standard uncertainty of 0.55% (Hülsen et al., 2016). Several small power lamps are used in a portable calibration facility to provide this traceability to the measurements of QASUME when it is measuring outside of of the optical laboratory of PMOD/WRC. At several occasions, this calibration chain was validated by moving QASUME to the PTB and directly measuring the spectral irradiance of the high temperature blackbody (Gröbner and Sperfeld, 2005; Hülsen et al., 2016). The latest validation campaign was performed in March and April 2022 to validate the spectral irradiance measurements of QASUME and QASUME-IR for the extended spectral range from 300 nm to 1700 nm. Further to the measurements in front of the high temperature blackbody, also irradiance measurements using the tuneable laser facility TULIP were performed with both spectroradiometers. The small power lamps in the portable calibration facility of QASUME provided the linkage between the PMOD/WRC irradiance scale and the irradiance measurements at the two facilities of the PTB.

The spectral irradiance measurements at PTB were performed with QASUME and QASUME-IR using the irradiance scale realised at PMOD/WRC from the average of seven 1 kW FEL transfer standard lamps, which were in turn calibrated relative to the national primary standard of PTB for spectral irradiance, the high temperature blackbody BB3200pg at different times since 2003. Therefore this so called QASUME irradiance scale represents a long-term average of the spectral irradiance scale realised by PTB. Figure 1 shows the ratio of spectral irradiances of the blackbody and the irradiances at specific wavelength settings of TULIP measured with QASUME and QASUME-IR relative to the irradiances of these 2 (spectral) irradiance sources. As can be seen in the figure, the measurements, with the exception of a few points measured at TULIP, are all within the expanded uncertainties of the spectral irradiance measurements of QASUME and QASUME-IR, thus validating the spectral irradiance scale at PMOD/WRC, and specifically the spectral irradiance measurements of QASUME and QASUME-IR when deployed outside of PMOD/WRC.

# 4 Validation of TSIS-1 HSRS solar spectrum

The retrieval of atmospheric transmission and aerosol optical depth from ground-based spectral direct solar irradiance measurements uses the Beer-Lambert law,

$$I_\lambda = I_{0\lambda} R_{\mathrm{SE}} e^{-\tau_\lambda m}, \tag{1}$$

where $I_\lambda$ represents the solar irradiance measured at wavelength $\lambda$, $I_{0\lambda}$ the solar irradiance at the top of the atmosphere, $R_{\mathrm{SE}}$ the sun-earth distance normalised to 1 AU, $\tau_\lambda$ the total optical depth and $m$ the airmass. The zero airmass extrapolation procedure as described in Gröbner et al. (2017) is used, by performing a linear regression of the logarithm of the spectral solar irradiance measurements with respect to airmass to retrieve the spectral solar irradiance value at airmass 0, representing the ToA solar irradiance at wavelength $\lambda$. This procedure assumes that during the duration of these measurements the atmospheric





**Figure 1.** Ratio of the spectral irradiances obtained from TULIP (blue dots) and of the high temperature blackbody BB3200pg (red line) with QASUME (300 nm to 550 nm) and QASUME-IR (550 nm to 1700 nm) using as reference 1kW FEL transfer standards traceable to the SI (QASUME scale) with respect to the spectral irradiances provided by the PTB. The gray shaded area represents the expanded relative uncertainty of the spectral irradiance measurements of QASUME and QASUME-IR.

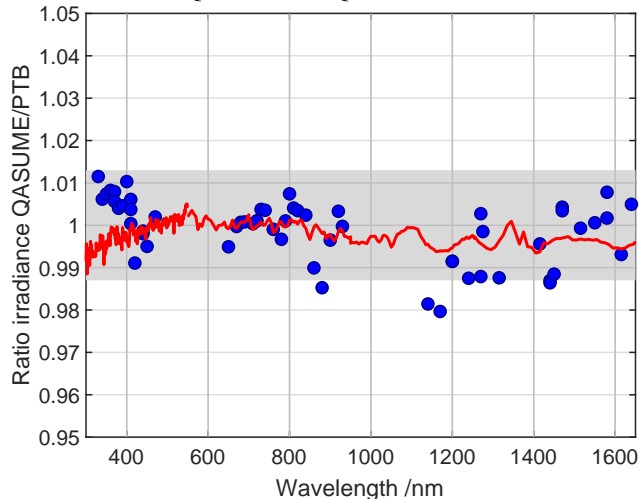

transmission $\tau_\lambda$ remains constant, and that any remaining small atmospheric variations are uncorrelated and therefore random from one day to the next. Furthermore, at wavelengths shorter than 310 nm, the extrapolated solar ToA spectra from the
QASUME spectroradiometer are bandwidth corrected following the procedure described in Gröbner et al. (2017).

From the available measurements in the period 6 to 22 September 2022, between 8 and 15 half-days were selected from the data set of each spectroradiometer, to retrieve ToA solar spectra for each half day period by zero airmass extrapolation. Subsequently, the mean ToA solar spectrum and the corresponding standard error were computed from these retrievals for each spectroradiometer. Figure 2 shows the extrapolated ToA solar spectra in the upper figure and the relative standard error of
the mean in the bottom. The ToA solar spectrum from TSIS-1 HSRS, convolved with a nominal 1 nm FWHM triangular line spread function, is also shown in the upper part of the figure.

The larger variabilities seen in Figure 2 are from spectral regions with strong atmospheric gas absorption where the atmospheric variability is higher. These spectral regions are also visible in the upper figure as remaining absorption features in the extrapolated ToA solar spectra, relative to the TSIS-1 HSRS solar spectrum. This also underlines once more the limitations
inherent in the ground-based zero airmass extrapolation methodology, since in these spectral regions with strong atmospheric absorption, the zero airmass extrapolation using the Beer-Lambert law fails due to the assumption of constant atmospheric absorption and unsaturated trace gas absorptions.

The ratio between the extrapolated ToA solar spectra shown in Figure 2 with the TSIS-1 HSRS spectrum, convolved with the respective line spread functions of the spectroradiometers, is shown in Figure 3. A 10 nm moving average is applied to the
ratios to remove some of the high frequency spectral variability. The agreement between the extrapolated ToA solar spectra

**Figure 2.** Top: Top of atmosphere solar spectra derived from zero airmass extrapolations by QASUME (blue), QASUME-IR (red), BTS (yellow), and PSR (violet) and TSIS-1 HSRS (green) convolved with a 1 nm FWHM triangular line spread function. The gray circles represent the solar irradiances measured with PFR N01. Bottom: Relative standard error of the mean of the ToA solar spectra shown in the top figure using the same color scale.

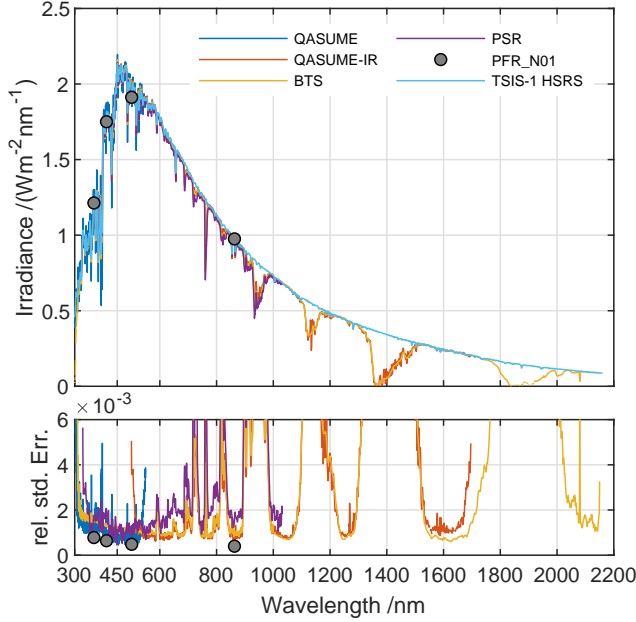

and TSIS-1 HSRS in the spectral regions unaffected by strong gas absorptions are mostly within 1%. The slightly larger deviations seen with the array spectroradiometers PSR and BTS in the ultraviolet spectral region below 400 nm, likely comes from unaccounted wavelength errors and line spread function related artifacts when convolving the high spectral resolution spectrum TSIS-1 HSRS with the spectrally varying line spread functions of the PSR and BTS spectroradiometers. Table 1 lists 235 the relative differences between the spectroradiometers and TSIS-1 HSRS for these selected spectral regions.

The consistency between TSIS-1 HSRS and the ToA solar irradiances retrieved with PFR N01 is also excellent, with relative differences below 1%, thus confirming the results already published in Kouremeti et al. (2022).

The spectral measurements with the lowest uncertainties and with the best agreement are obtained with the QASUME (300 nm to 550 nm), and QASUME-IR (550 nm to 1700 nm) scanning spectroradiometers. As can be seen in the figure, the 240 agreement of the ToA solar spectra retrieved with QASUME and QASUME-IR agrees within 1% with the TSIS-1 HSRS solar spectrum in the spectral regions unaffected by strong gas absorption. This agreement is well within the combined uncertainties of the instruments, and confirms the excellent agreement observed between measurements of QASUME in the 2016 campaign and TSIS-1 HSRS (Coddington et al., 2021).

Metrological traceability of space-based solar irradiance measurements is currently not achieved. While pre-flight calibra-245 tions can be made traceable, the harsh environment of space following the shock of launch means that generally, radiometric ground calibrations cannot be relied upon in space (Fox and Green, 2020). The mission TRUTHS, Traceable Radiometry Un-



**Figure 3.** Spectral ratio of the zero airmass extrapolated ToA solar spectra from QASUME (blue), QASUME-IR (read), PSR (violet), and BTS (yellow) relative to the TSIS-1 HSRS solar spectrum convolved with the respective line spread functions of the spectroradiometers. The coloured lines are a 10 nm moving average of the spectral ratios in spectral regions with no or only weak atmospheric trace gas absorption, while the gray line represents the full spectral ratio of BTS to TSIS-1 HSRS. The gray circles represent the ratios of the spectral solar irradiances of PFR N01 with TSIS-1 HSRS convolved with the spectral filter transmissions of PFR N01.

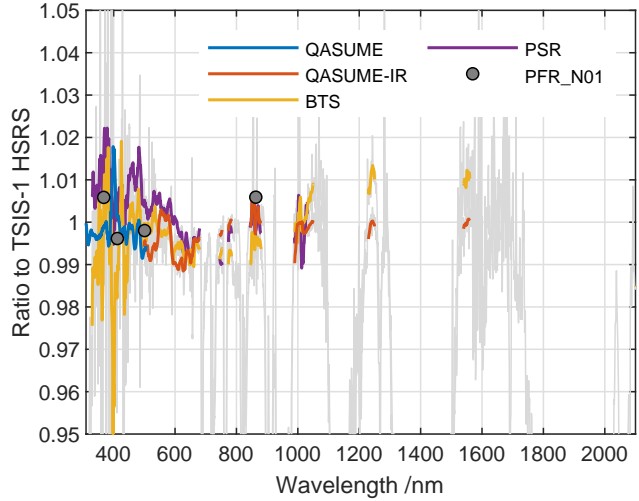

**Table 1.** Ratios of ToA solar spectra from QASUME, QASUME-IR, BTS and PSR relative to TSIS-1 HSRS averaged over different spectral intervals unaffected by strong trace gas absorption features. The values in parenthesis represent the standard deviation of the spectral ratio within this spectral interval.

| Spectral region /nm | | Ratio to TSIS-1 HSRS | | | |
|---|---|---|---|---|---|
| Min | Max | QASUME | QASUME-IR | BTS | PSR |
| 300 | 400 | 0.999(8) | | | |
| 400 | 550 | 0.998(7) | | 0.998(17) | 1.006(4) |
| 550 | 680 | | 0.995(5) | 0.994(5) | 0.998(4) |
| 745 | 753 | | 0.996(2) | 0.991(3) | 0.991(1) |
| 775 | 784 | | 0.999(2) | 0.990(3) | 0.999(1) |
| 845 | 880 | | 1.002(8) | 0.995(7) | 1.001(1) |
| 990 | 1050 | | 0.999(5) | 1.002(6) | 0.998(1) |
| 1230 | 1250 | | 1.000(2) | 1.011(4) | |
| 1540 | 1560 | | 0.994(15) | 1.002(15) | |

derpinning Terrestrial- and Helio- Studies (TRUTHS), expected to be launched at the end of this decade, will, for the first time, establish high-accuracy SI traceability in orbit. As stated in Fox and Green (2020), the direct use of a primary standard and

**Figure 4.** Relative standard uncertainty of the ToA solar spectrum from TSIS-1 HSRS (red) and from QASUME and QASUME-IR (blue).

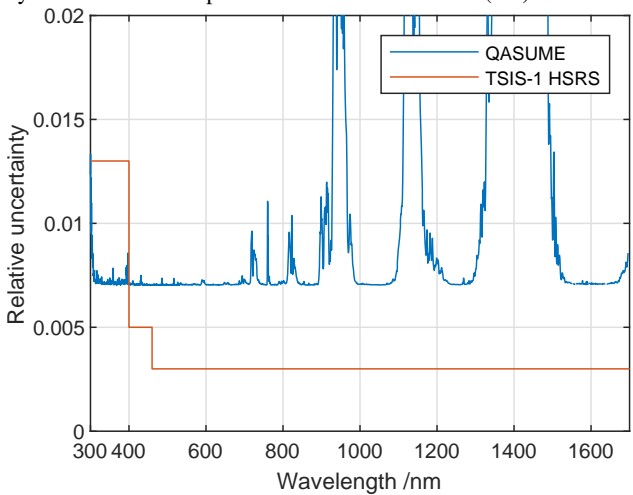

replication of the terrestrial traceability chain will extend the SI into space, in effect realizing a metrology laboratory in space,
providing and enabling SI-traceable measurements of unequivocal accuracy.

The comparison of the TSIS-1 HSRS ToA solar spectrum with ToA solar irradiance measurements obtained from extrapolated SI traceable ground-based measurements provides such a metrological validation. Therefore, this comparison also establishes a direct traceability link to the SI for the relevant spectral regions of the TSIS-1 HSRS ToA solar spectrum via the ground based measurements discussed in this study.

Furthermore, the validation of TSIS-1 HSRS with the extrapolated ToA solar spectrum from QASUME/QASUME-IR can also be used to confirm, and even reduce the uncertainty of TSIS-1 HSRS in the spectral regions where the uncertainties of TSIS-1 HSRS are larger than the ones of the QASUME/QASUME-IR ToA solar spectrum. Figure 4 shows the standard relative uncertainties as given for TSIS-1 HSRS (Coddington et al., 2021), and the ones of the QASUME/QASUME-IR ToA solar spectrum. The latter are obtained by combining the measurement uncertainties of 0.7% (see section 2.1) with the standard error of the ToA retrieval shown in the bottom of Figure 2. While the uncertainties of TSIS-1 HSRS are lower than those of QASUME for wavelengths longer than 400 nm, at shorter wavelengths the uncertainty of TSIS-1 HSRS can be significantly reduced from 1.3% to 0.8%, which is the relative standard uncertainty of the QASUME ToA solar spectrum. This reduced uncertainty of TSIS-1 HSRS is valid for the spectral range starting at 308 nm until 400 nm, at which point the TSIS-1 HSRS uncertainty drops to 0.5%.

## 5  Spectral Aerosol Optical Depth

The spectral optical depth $\tau_a$ is derived by rearranging equation 1 and expanding the optical depth $\tau$,

$$\tau_a(\lambda) = (\log \frac{I_0(\lambda)}{I(\lambda)} - \sum_{i=1}^{N} \tau_i(\lambda)m_i)/m_a - \tau_R(p,\lambda) \qquad (2)$$





**Figure 5.** Spectral optical depth of the atmosphere between 300 nm and 2100 nm obtained from direct spectral solar irradiance measurement on 10 September 2022 at 8:29 UTC. The gray bars represent the spectral channels used by the CIMEL and PFR sunphotometers to retrieve the AOD.

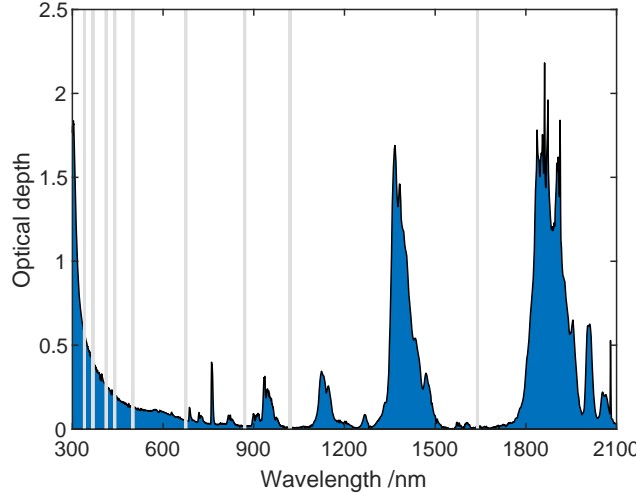

where the $\tau_i$ represent the optical depths of the atmospheric trace gas $i$ and $m_i$ the corresponding airmass; $\tau_R(p, \lambda)$ stands for the Rayleigh scattering term which depends on the atmospheric pressure $p$ with $\lambda$ the corresponding wavelength. The Sun-Earth

correction factor $R_{SE}$ has been omitted for convenience.

For an accurate AOD retrieval the influence of trace gas absorptions $\tau_i$ need to be known, so usually, the AOD is retrieved only in the spectral regions where the uncertainty due to their influence is small, either because the absorption is negligible or because it can be corrected for. This is for example the case for atmospheric ozone, which needs to be subtracted in the short ultraviolet below 340 nm, or in the Chappuis band between approximately 450 nm and 750 nm.

The airmasses $m_i$ depend on the vertical distribution of the respective trace gas in the atmosphere; apart from ozone, which is mostly located in the stratosphere, all other trace gases are assumed to be well mixed and distributed in the troposphere. Since this information is usually not known, for simplicity it is assumed to be equal to the aerosol airmass $m_a$, which is a reasonable approximation as long as the solar zenith angle is below $75°$.

The spectral range under investigation, 300 nm to 2100 nm, contains spectral absorption features from several atmospheric

trace gases, such as water vapour ($H_2O$), ozone ($O_3$), $CO_2$, $CH_4$, $NO_2$, and several additional oxygen absorption bands, the most prominent being at 762 nm. Figure 5 shows the optical depth retrieved from a solar irradiance measurement on 10 September 2022 at 8:29 UTC.

Several prominent absorption features are visible in the figure, stemming from ozone in the ultraviolet and around 600 nm, several water vapour bands, as well as $CO_2$ close to 1600 nm. The AOD is retrieved in the spectral regions with minimum trace

gas absorption, which is where the filter channels of the sunphotometers are located, as indicated in the figure by the gray bars.

For the measurements described in this study, the only trace gas that was taken into account was ozone, since the absorption from the other trace gases is negligible in these spectral regions, with the exception of the IR channel at 1640 nm, as discussed



later. This is also consistent with the AOD retrieved by the GAWPFR sunphotometer that will be used to validate the spectral AOD measurements. The AOD retrieved from the CIMEL sunphotometer is based on the AERONET procedures and is

corrected for several trace gases, as described in Holben et al. (1998). However, apart from the 1640 nm channel, the optical depth corrections from $NO_2$ and $H_2O$ in the corresponding channels of the CIMEL sunphotometer are less than 0.002 for the conditions of Izaña.

## 5.1 Uncertainty budget

The uncertainty estimation of the spectral AOD is based on equation 2 and is shown here for the QASUME and QASUME-IR

spectroradiometers. A similar uncertainty estimation can also be done for the PSR and BTS spectroradiometers (not shown).

### 5.1.1 Atmospheric transmission

The most important uncertainty contribution arises from the atmospheric transmission term $\log \frac{I_0}{I}$. When $I_0$ is obtained from zero airmass extrapolation, then the uncertainty is essentially equal to the standard error of the mean of a set of zero airmass extrapolations (Toledano et al., 2018). For solar irradiance measurements traceable to the SI, combined with an independent

ToA solar spectrum, the combined uncertainty is significantly larger. For QASUME and QASUME-IR, the corresponding uncertainty is obtained from the spectral solar irradiance relative measurement uncertainty of 0.7%, combined with the uncertainty of the TSIS-1 HSRS ToA solar spectrum. As stated in Coddington et al. (2021), the standard relative uncertainty of the TSIS-1 HSRS ToA solar spectrum is 1.3% below 400 nm, 0.5% between 400 nm and 460 nm, and 0.3% between 460 nm and 2400 nm. As discussed in the previous section, the uncertainty of TSIS-1 HSRS can be reduced in the spectral interval 310 nm

to 400 nm to 0.8%.

### 5.1.2 Ozone and Rayleigh optical depth

The additional uncertainty contributions coming from the Rayleigh and ozone optical absorption terms are much smaller, especially at longer wavelengths. The uncertainty in the ozone absorption is obtained by assuming a default uncertainty in the total ozone column of 1%, as obtained from the measurements of QASUME itself (Egli et al., 2022). Assuming a representative

total ozone column value for this study of 280 DU, this yields an uncertainty of 0.007 at 310 nm, but less than 0.001 in the spectral bands measured by the sunphotometers. This implies that even if the sunphotometers used different data sources for their total ozone column correction, the expected uncertainties in that correction are negligible.

Similarly, the Rayleigh scattering is calculated using the formula by Bodhaine et al. (1999). Since this formula has been accepted as the standard by all aerosol monitoring networks, the only relevant uncertainty term is related to the atmospheric

pressure, which is measured at the Izaña observatory with an uncertainty of 1 mbar and a correspondingly negligible contribution to the overall uncertainty.

**Table 2.** Uncertainty budget of AOD for the QASUME and QASUME-IR spectroradiometers at the standard spectral channels of the PFR and CIMEL sunphotometers for an airmass of 1.5. The channel values in parenthesis represent the central wavelengths of the PFR, while the last entry at 1560 nm has the corresponding channel of the CIMEL sunphotometer at 1640 nm.

| | Relative std. uncertainty in % | Std. uncertainty |
|---|---|---|
| Channel /nm | $\log \frac{I_0}{I}$ | optical depth |
| 310 | 1.06 | 0.014 |
| 340(368) | 1.06 | 0.0091 |
| 412(440) | 0.86 | 0.0077 |
| 500 | 0.76 | 0.0071 |
| 675 | 0.76 | 0.0071 |
| 870 | 0.76 | 0.0071 |
| 1020 | 0.76 | 0.0071 |
| 1560 | 0.76 | 0.0071 |

### 5.1.3 Other trace gases

As stated previously, the contribution from other trace gases are negligible in the spectral regions selected by the sunphotometers, with the exception of the IR-channel at 1640 nm, where water vapour, $CO_2$ and $CH_4$ contribute significantly to the absorption optical depth. Following the calculations described in Giles et al. (2019), this yields a correction of 0.0125 for the conditions at this measurement site. However at the nearby spectral region centered at 1560 nm, these trace gas absorptions are much reduced, which is therefore where the AOD for QASUME-IR and the BTS are retrieved. The uncertainty budget for spectral AOD retrieved by QASUME/QASUME-IR is shown in Table 2. For convenience, it is shown for the standard spectral channels of the GAWPFR and AERONET sunphotometers as well as at 310 nm. A constant uncertainty contribution of 0.002 is added to take into account minor absorptions arising from unaccounted trace gas absorbers, while at 310 nm the ozone uncertainty increases that component to 0.007.

The high-resolution measurements provided by the FTIR instrument were used to test the validity of the gas absorption assumptions in the near infrared region. Figure 6 shows an example atmospheric optical depth between 1500 nm and 1700 nm, as determined from the gradient of the Langley fit to solar FTIR data taken on the 22nd September 2022. In addition to the high resolution data (black line), three spectrally broadened results are shown with the FWHM values of 2.4 nm, 4 nm corresponding to the spectral resolution of the QASUME-IR and BTS spectroradiometers respectively. Table 3 shows the optical depth difference between the broadened results and the baseline optical depth from the high resolution data at the standard sunphotometer spectral channels in the near-infrared region (1020 nm and 1640 nm) and the alternative channel at 1560 nm identified for the spectroradiometers. The baseline value is determined as the mean value of the high resolution data points within the respective bandwidths that are unaffected by gas absorption lines. The reported uncertainty is the standard deviation of the optical depth differences seen at each wavelength across the FTIR Langley analyses over the campaign.





**Figure 6.** Example of high resolution atmospheric optical depths determined from Langley analysis on 22nd September 2022 (black line). Broadened results for spectral resolutions of 2.4 nm, 4 nm and 6 nm are also shown (red, blue and green lines respectively)

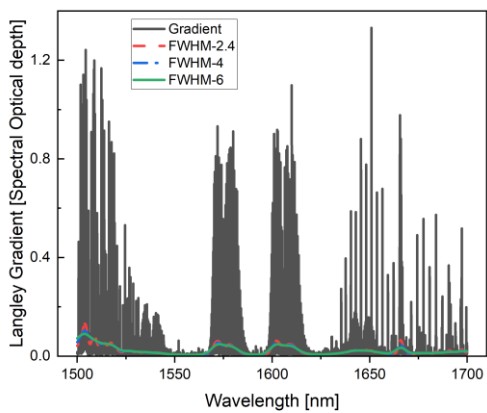

**Table 3.** Optical Depth differences between high resolution baseline and broadened spectra

| Center Wavelength /nm | Optical depth difference at a spectral resolution of | | |
|---|---|---|---|
| | 2.4 nm | 4 nm | 6 nm |
| 1020 | 0.00126(67) | 0.00116(62) | 0.00117(63) |
| 1560 | 0.00065(67) | 0.00076(69) | 0.00085(69) |
| 1640 | 0.0139(20) | 0.0127(18) | 0.0121(18) |

The results at 1640 nm confirm that the 0.0125 optical depth correction applied at this wavelength is appropriate given the gas contributions actually observed during the measurement campaign, while the results at the other wavelengths indicate that the constant 0.002 trace gas uncertainty contribution is a conservative estimate of the uncertainty in the near-infrared region.
Table 3 also shows that these assumptions remain valid for spectral resolutions up to FWHM of 6 nm.

## 5.2 AOD comparison

The spectral AOD from the cloudfree days of the campaign was retrieved from the QASUME/QASUME-IR, BTS and PSR spectroradiometers, and compared to 2 sunphotometers from the GAWPFR and AERONET network. In total 10 days could be used for the comparison where all instruments provided AOD for most or all of the day. The data from the periods where
instruments malfunctioned or were calibrated have been removed from the datasets.

As an example, Figure 7 shows the spectral AOD from 13 September 2022, at 16:00 UTC.

The overall agreement of 0.01 in AOD between the instruments is remarkable, considering, the very low spectral AOD at this site. The larger spectral variability of the AOD from the PSR and BTS systems below 400 nm arises from convolving the high



**Figure 7.** Spectral Aerosol Optical Depth from QASUME (blue), QASUME-IR (red), BTS (yellow), PSR (violet) from measurements obtained on 13 September 2022 at 16:00 UTC. The circles are from the PFR (blue), and CIMEL (green) sunphotometers. The gray curve represents the complete spectral optical depth over the whole spectral interval measured by the spectroradiometers, including spectral regions also contaminated by unaccounted trace gas absorptions.

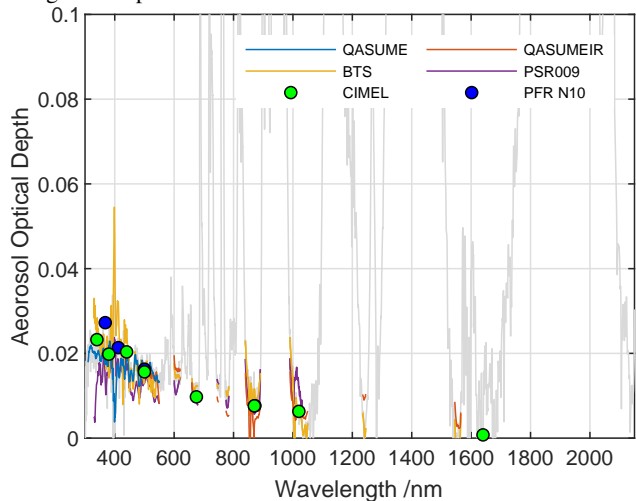

spectral resolution TSIS-1 HSRS solar spectrum with the line spread functions of the spectroradiometers, which is probably

due to the difficulty in accounting for the variable resolution of these instruments in this spectral region. On the other hand, the spectral variability of the AOD from QASUME is less than 0.005 (peak to peak), with the exception of a dip of about 0.01 at 400 nm which is currently unexplained.

While predefined narrowband filters, once manufactured and assembled, do not permit switching to another spectral region, this is the clear advantage of measuring the complete AOD spectrum, as shown in Figure 7. This is clearly seen for the IR filter

at 1640 nm, where the trace gas absorption from $CO_2$, $CH_4$, and $H_2O$ accounts for about 70% of the total absorption in this spectral region, while on the other hand, a spectral interval with minimal trace gas absorption centered at 1560 nm is available, which can be used to assess the AOD in the infrared region.

Figure 8 shows the diurnal variation of the AOD at the common spectral channels of the sunphotometers for this same day. For the PFR, the spectral AOD at 340 nm and 440 nm was obtained using the Ångstrom coefficients derived from its 4

spectral channels. The spectral AOD of the spectroradiometers were averaged over a 5 nm wide spectral band centered at these wavelengths. The gray shaded area represents the WMO accepted uncertainty coverage interval in which 95% of the data from different instruments should be located, in order for them to be considered equivalent (WMO, 2005). The reference instrument considered for the WMO criterium is the PFR N10 at all channels shorter than 1640 nm, while at this particular channel the CIMEL radiometer is used instead.

Considering the very low AOD on this day, the agreement between the instruments is excellent and well within the WMO criterium. Especially noteworthy is that the AOD from the two filter radiometers is obtained using the zero airmass extrapolation



**Figure 8.** Diurnal variation of the AOD on 13 September 2022 for the spectral channels of the CIMEL sunphotometer. The AOD from QASUME and QASUME-IR (blue dots), BTS (light blue) and PSR (yellow) are averaged over a 5 nm wide spectral band centered on the respective wavelength. The AOD from the PFR (orange) shown at 340 nm, 380 nm, and 440 nm were interpolated to the nearby spectral channels of the CIMEL sunphotometer (green) using the Ångstrom coefficients retrieved from its 4 spectral channels. The measurements of QASUME-IR and BTS at nominally 1640 nm were obtained from averaging their measurements at 1560 nm to avoid the trace absorptions at 1640 nm. The gray area represents the WMO limit in which the measurements between different instruments are assumed equivalent.

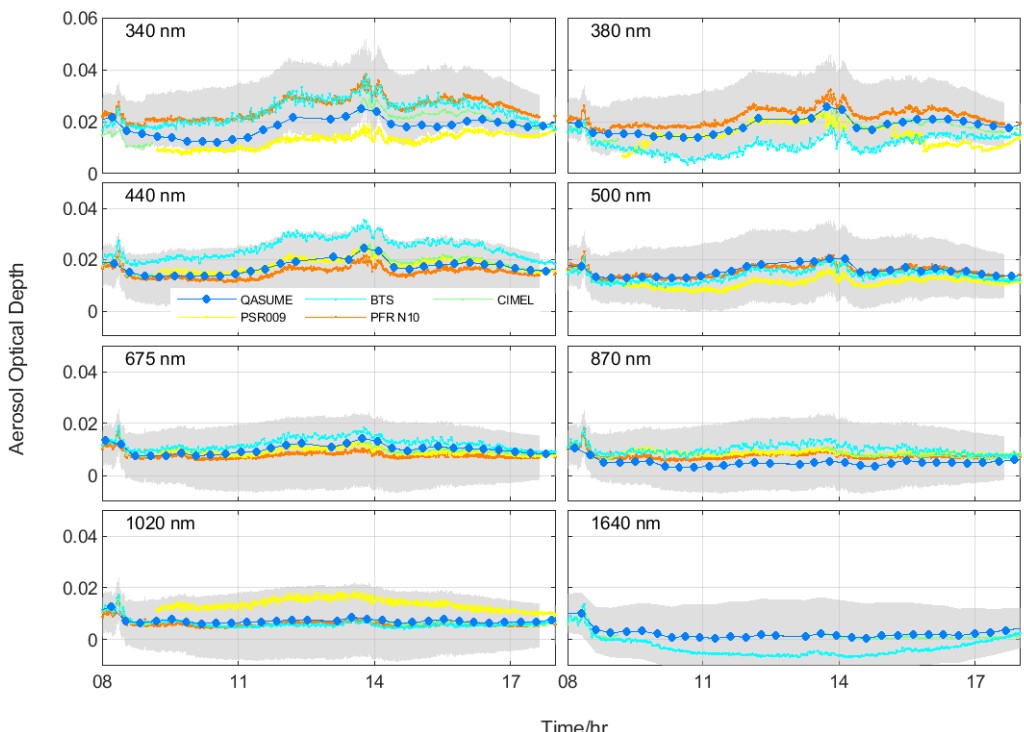





**Table 4.** Average AOD differences and their standard deviations from the following cloudfree days: 6, 7, 8, 9, 10, 13, 14, 15, 17, and 22 September 2022 using QASUME, QASUME-IR, BTS, PSR, and CIMEL relative to the PFR for the selected spectral channels of the sunphotometers. The comparison at 1640 nm is done relative to the CIMEL since the PFR does not have a corresponding channel in this spectral range. The bold values indicate that for this particular spectral channel and instrument 95% or more of the data is within the WMO limits with respect to the reference instrument.

| | AOD difference to PFR (CIMEL at 1640 nm)·$10^3$ | | | | | | | |
| Instrument | 340 nm | 380 nm | 440 nm | 500 nm | 675 nm | 870 nm | 1020 nm | 1640 nm |
| --- | --- | --- | --- | --- | --- | --- | --- | --- |
| QASUME | **-7.8(52)** | **-3.9(43)** | **-0.8(35)** | **-1.2(35)** | | | | |
| QASUME-IR | | | | | **4.3(28)** | **-3.2(18)** | **2.6(23)** | **3.2(17)** |
| BTS | **-0.6(35)** | **-8.5(57)** | 7.6(36) | **-1.5(18)** | **4.8(22)** | **4.6(27)** | **-0.2(16)** | **-4.0(33)** |
| PSR | **-0.8(57)** | **-6.5(47)** | **3.1(21)** | **-0.8(25)** | 7.1(47) | **1.7(41)** | 7.6(49) | |
| CIMEL | **-3.0(27)** | **-1.8(18)** | **2.5(22)** | **-0.7(12)** | **2.1(15)** | **0.6(13)** | **0.2(11)** | |

from this measurement period, while the AOD from the spectroradiometers are based on solar irradiance measurements and an independent solar ToA solar spectrum. One should note that the slightly negative AOD values by the BTS at 1640 nm only arise due to the extremely low AOD at this wavelength; while being clearly unphysical, they are still within 0.01 of the CIMEL radiometer and QASUME-IR and thus within the WMO criterium.

Figures S1 to S9 in the supplement to this paper show the diurnal variation for the other cloudfree days of the intercomparison.

    The complete datasets of spectral aerosol optical depth from the 10 cloudfree days were also compared to the PFR at all channels below 1640 nm, while for this channel the CIMEL radiometer was used. Table 4 provides a summary of this

comparison as averages and standard deviations for all instruments and at all spectral channels. The bold values shown in the table also indicate that the datasets meet the WMO criterium, e.g., 95% of the available data is within the WMO limit as defined previously. As can be seen in the table, the agreement is excellent, with average differences below 0.01 for all instruments and at all spectral channels (which is also the reported AOD uncertainty of the PFR and CIMEL sunphotometers). Furthermore, the WMO criterium is met in all but three cases, demonstrating that the spectroradiometers and the reference radiometers meet

the WMO equivalency criterium at nearly all spectral channels.

    A further comparison between the AOD retrieved from the QASUME and QASUME-IR spectroradiometers and a number of narrowband filter radiometers was performed during the 5th filter radiometer comparison (FRC-V) held at PMOD/WRC in Davos, Switzerland from 27 September to 25 October 2021. This comparison occurred prior to the campaign presented here, and produced similarly consistent results with the reference sunphotometers deployed at the FRC-V (Kazadzis et al., 2023).

The FRC-V also indicated that the AOD derived from solar spectra traceable to the SI are consistent with the AOD derived by reference sunphotometers and are within the WMO limits when using well calibrated spectroradiometers and an accurate ToA solar spectrum, such as TSIS-1 HSRS, or QASUMEFTS. The FRC-V was additionally interesting because the measurements



took place at a measurement site at mid-latidues and not at one of the main high altitude calibration sites where most of these reference filter radiometers were calibrated using zero airmass extrapolations.

## 6  Conclusion

Spectral measurements of direct solar irradiance traceable to the SI were performed by several spectroradiometers during a three week campaign at the Izaña Atmospheric Observatory (IZO) located on the island of Tenerife, Canary Islands, Spain. The QASUME and QASUME-IR spectroradiometers were calibrated daily using small power lamps and a portable calibrator to provide traceability to the SI and ultimately to the high temperature blackbody of the PTB, while the PSR was calibrated before and after the campaign in the laboratory of PMOD/WRC. The BTS spectroradiometers were calibrated partly at the PTB and at the ISO 17025 calibration laboratory of Gigahertz Optik GmbH. During the campaign the system was checked with a portable calibrator.

The spectroradiometers measured direct solar spectral irradiance in the spectral range 300 nm to 550 nm (QASUME), 550 nm to 1700 nm (QASUME-IR), 300 nm to 2100 nm (BTS), and 316 nm to 1030 nm (PSR), with relative standard uncertainties of 0.7%, 0.95%/2.1% and 0.9% for QASUME/QASUME-IR, BTS (below/above 1000 nm) and PSR respectively. The uncertainty budget of QASUME/QASUME-IR was validated prior to this campaign at the PTB by measuring the spectral irradiances of the high temperature blackbody BB3200pg and from the tuneable laser facility TULIP and comparing them to the spectral irradiance scale obtained in the laboratory of PMOD/WRC using a set of transfer standard tungsten halogen lamps traceable to the same high temperature blackbody.

The Top of Atmosphere solar irradiance spectrum was retrieved from the solar irradiance measurements using zero airmass extrapolation during cloudfree conditions and compared to the TSIS-1 HSRS solar spectrum. The agreement between the extrapolated ToA solar spectra and TSIS-1 HSRS was excellent and well within the combined uncertainties of the spectroradiometers over the full spectral range. Especially the measurements of QASUME/QASUME-IR between 300 nm and 1700 nm were within 1% (peak to peak) of the TSIS-1 HSRS Solar irradiance spectrum in the spectral regions not affected by trace gas absorptions. Based on this comparison, the measurements of QASUME could be used to reduce the stated relative standard uncertainty of the TSIS-1 HSRS ToA solar spectrum in the spectral range 308 nm to 400 nm from 1.3% to 0.8%.

The spectral AOD retrieved from the solar measurements of the spectroradiometers using TSIS-1 HSRS as the reference solar spectrum for the Top of Atmosphere were compared to the AOD from two reference narowband filter radiometers belonging to the GAWPFR and AERONET networks. The agreement in nearly all common spectral channels of the radiometers was well within the WMO acceptance limits defined by the WMO.

These measurements demonstrate that it is now possible to retrieve spectral AOD over the extended spectral range from 300 nm to 1700 nm using solar irradiance measurements traceable to the SI using laboratory calibrated spectroradiometers with similar quality as from traditional Langley-based calibrated instruments. The main improvement to previous investigations is the recent availability of the high spectral resolution TSIS-1 HSRS solar spectrum with very low uncertainties which provides



the Top of Atmosphere reference for the spectral atmospheric transmission measurements obtained from ground based solar
irradiance measurements.

While the spectral AOD retrieved from traceable spectral solar irradiance measurements of the spectroradiometers has been
shown to be in excellent agreement with the AOD from AERONET and GAWPFR sunphotometers, the corresponding uncertainty is still slightly higher than the latter due essentially to the uncertainty of the spectral atmospheric transmission measure-
ment resulting from the combined uncertainty of the ToA solar spectrum and of the ground-based measurements.

Nevertheless, the significant improvement of this study is that AOD can now be retrieved from solar irradiance measurements fully traceable to the SI. Furthermore, this process does not rely anymore on Langley-based calibrations of reference radiometers performed at high altitude sites with consequent loss of traceability after their relocation to their network calibration site.

*Data availability.*   The main datasets used in this study are available at the following community link on Zenodo, https://zenodo.org/communities/19env04-mapp, and specifically at https://doi.org/10.5281/zenodo.8043872.

*Author contributions.*   JG analysed the QASUME and BTS datasets and wrote the manuscript, NK operated the PSR and PFRs and analysed their data, RZ and MR operated the BTS, GH operated QASUME & QASUME-IR, SN, PS, PS, and KS calibrated QASUME, QASUME-IR, BTS, and PSR at PTB, AB provided data from the CIMEL and organised the campaign, SK supported the PFR analysis and provided
comments to the manuscript, TG, KM, DM, MC operated the FTIR data and wrote the relevant section.

*Competing interests.*   At least one of the (co-)authors is a member of the editorial board of Atmospheric Measurement Techniques.

*Acknowledgements.*   This work has been supported by the European Metrology Program for Innovation and Research (EMPIR) within the joint research project EMPIR 19ENV04 MAPP "Metrology for aerosol optical properties". The EMPIR is jointly funded by the EMPIR participating countries within EURAMET and the European Union. The authors would also like to thank the staff of the Izaña atmospheric
Observatory for their support during the campaign and for providing the ancillary data used in the analysis.





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
