# Peer review of "Spectral Aerosol Optical Depth from SI-traceable Spectral Solar Irradiance Measurements"

_Atmospheric Measurement Techniques, 2023_

## Referee Comment (RC2)

Spectral Aerosol Optical Depth from traceable Spectral Solar Irradiance Measurements to the SI
By J. Grübner, N. Kouremeti, et al.
https://doi.org/10.5194/amt-2023-105

This article presents the results of an intensive ground-based field campaign of:
1) spectroradiometric measurements of direct solar irradiance using several systems with SI-traceable laboratory calibrations (and their inter-comparison),
2) the interpolation to zero air-mass to derive a top-of-atmosphere solar irradiance using the Langley method and subsequent comparison to and validation of a hybrid solar spectral irradiance spectrum from high accuracy satellite observations that has recently been recommended as a new reference standard,
3) the retrievals and intercomparisons of aerosol optical depth from the spectroradiometers and two "master" sunphotometers from global observing networks.

The article is well-written and comprehensive in establishing first-ever demonstration of SI-traceable spectral aerosol optical depth retrievals using a calibration chain of spectroradiometers that is traceable to SI. Furthermore, the AOD retrieved from the spectroradiometers was shown to compare to within acceptance limits to "master" reference sunphotometers of key global networks (AERONET and GAWPFR) to within World Meteorological Organization (WMO) criteria of +/- 0.01 AOD units. The measurement campaign is described in detail as are the radiometric calibrations of the individual sensors and the description of the methodologies used in the inter-comparisons. I believe this article presents research results that are of high interest to the Earth science community.

I have a few comments below that I hope will improve the paper beyond it's already excellent presentation.

1. I agree that the positive comparisons between the derived top-of-atmosphere solar spectral irradiance from the spectroradiometers, particularly the QASUME (as expected from previous findings) and QASUME-IR, and the TSIS-1 HSRS are wonderful. To have your data independently validate the TSIS-1 HSRS at the ~1% level from 300-1700 nm is a very important outcome of this study, and something I will communicate to users of the TSIS-1 HSRS reference spectrum. I have been eagerly anticipating the results of this intercomparison experiment since you and Natalia told me of it at the most recent IRC symposium. As you mention in the paper, uncertainties in previous satellite solar spectra and that derived from ground-based instruments have been too large to ever show this self-consistency before.
   a. In a couple of places in the paper, you hypothesize the somewhat larger differences below 400 nm between the TSIS-1 HSRS and the PSR and BTS spectroradiometers could be due to lack of knowledge in the wavelength-dependent instrument profile functions of the PSR and BTS spectroradiometers. Essentially, as shown in many previous studies, due to the high solar structure in the UV, uncertainty in line profiles would certainty lead to more variability in a ratio of solar spectra. The TSIS-1 SIM instrument also has spectrally varying line profiles (published on the TSIS website at https://lasp.colorado.edu/tsis/data/ssidata/ ) and extensive ground calibrations measured line profiles at set points across the spectrum, with an instrument model then interpolating the line profile to a fine wavelength grid. Have you considered performing the convolutions in the reverse approach before taking the ratio (i.e. convolving the PSR and BTS instruments, and the HSRS spectrum, with the SIM line profiles that are known potentially to higher fidelity than the PSR and BTS spectroradiometers? Perhaps you would argue that extra effort is unnecessary given the positive comparison with the QASUME and QASUME-IR instrument results and I wouldn't disagree with that assessment, given that there is ample published literature that already exists showing positive UV spectral irradiance ratios with that degree of scatter (or even higher scatter).

b.  I am curious some of the differences shown in Figure 1 between QASUME and the PTB irradiances from the blackbody and TULIP. How are these comparisons done? Are they done in vacuum? Is some larger variability in comparing to TULIP attributed to laser stability, or in line profile of QASUME?

c.  Overall, I want to make sure I understand correctly how you used the FTIR data in this campaign. Was it only in verifying the corrections for trace gas optical depth as shown in Figure 6? Or did you also use the FTIR, in conjunction with QASUME and QASUME-IR to produce very high spectral resolution top of atmosphere solar irradiance like you have published in your previous paper on that topic? Meaning that the solar spectra shown in Figure 2 have already been convolved to the lower resolution of QASUME and QASUME-IR? I ask because I would be interested in obtaining your top-of-atmosphere, very high-resolution solar spectrum across the 300-1700 nm range for further validation of the solar lines in the hybrid HSRS spectrum, if it exists and you are willing to share.

2.  In general, I didn't understand the ultimate purpose for including the PSR and BTS spectroradiometers in the measurement campaign. I think this might just be because of my lack of familiarity with the AOD network and the long-term WMO calibration campaigns to maintain these networks. For example, do these instruments have additional importance because they have heritage in the longterm WMO calibration campaigns of AOD networks, new instrumentation for these purposes, or they demonstrate a relevant new commercial product that would replace some older instrumentation? I apologize if this has been answered in the paper by work you have cited and I didn't catch it.

3.  I understand why you discuss TRUTHS mission in context of establishing high-accuracy and SI-traceable accuracy in space. That is an important advance. However, I want to push back a little bit on the statement (line 244) that "Metrological traceability of space-based solar irradiance measurements is currently not achieved." As you show in this paper, the three-channel duty cycle approach to correcting on-orbit degradation of the TSIS-1 SIM, as captured by the TSIS-1 HSRS reference standard for a time period of >1.5 years of harsh solar exposure after TSIS-1 SIM launch in early 2018, has maintained those SIM ground calibrations as validated independently by QASUME and QASUME-IR to the ~ 1% level.  Would you consider adding the words "Direct and independent" at

the start of the sentence in quotes above, or further text as you see fit to provide context?

4. In section 5.1.2 about the uncertainty contributions in AOD retrievals due to ozone absorption (and correction), you discuss the sensitivity to AOD retrievals to total ozone column values. Has sensitivity to ozone cross section also been considered, or is there accepted "standard" ozone cross section values that are assumed by the AOD community?

5. I have general questions that are forward-looking and I wondered if you might consider addressing them in your conclusions. You discuss the significance that SI-traceable AOD values can now be retrieved from spectroradiometers, removing the reliance on standard Langley-based calibrations of reference radiometers at high-altitude sites and their subsequent relocation to network calibration sites. What are the next steps and what does this mean for AOD monitoring networks? Is there cost-savings with switching to the new method? Does it open up opportunities for higher-density AOD monitoring, or even monitoring from more challenging locations? Something else?

Minor Technical comments
Figure 2: Caption identifies TSIS-1 HSRS as green, but the legend gives it as a cyan color.
Figure 3: Caption should change "read" to "red"
Overall, the figures use inconsistent colors from one plot to the next for the different instruments. If not too much work, a consistent color usage would be nice.

---

## Author Comment (AC2)

Dear Odele,

Thank you very much for your comments and suggestions, which are very pertinent and will clarify our arguments.

See below our answers to the points you raised:

**1a)** We have done this reverse convolution you suggested using our previous results with the high resolution QASUMEFTS spectrum based on the campaign data from September 2016, which agrees well with the results you published in your TSIS-1 HSRS paper (Coddington et al., 2021, Figure 2). In contrast, here we are only using the medium-resolution datasets from our spectroradiometers, which have similar or even larger resolutions than the TSIS-1 SIM spectroradiometer, at least in the spectral range shorter than 400 nm. As you point out in 2), for the comparison between TSIS-1 HSRS, QASUME & QSAUME-IR are sufficient and BTS and PSR are not strictly needed for this argument, as QASUME & QASUME-IR have the lowest uncertainties.
Furthermore, also in response to your point 2), the PSR and BTS spectroradiometers are included in this study because they are commercial instruments in contrast to QASUME and QASUME-IR, and as we show, high fidelity spectral AOD can be retrieved with these calibrated instruments when using the TSIS-1 HSRS solar spectrum as reference.

**1b)** The comparisons are done at PTB in air, with the blackbody and TULIP facilities being in different buildings. Thus, we used small power lamp transfer standards to relate the measurements from both facilities. Since the spectral irradiance emitted from the blackbody is spectrally very smooth (Planck radiator with a radiation temperature around 3000 K), its measurement with QASUME and QASUME-IR is straightforward and was done within one day. In contrast, we spent several weeks at the TULIP facility to determine the spectral responsivity of QASUME at different central wavelengths, following two approaches:

1) We set TULIP to a specific wavelength and scan this laser wavelength with QASUME/QASUME-IR at fine spectral resolution as described in Hülsen et al., 2016, and we repeat this throughout the spectral range as shown in Figure 1. Unfortunately the QASUME-IR grating drive was found not to be sufficiently uniform at steps of 0.05 nm as required here, so we used a second approach described below:

2) We set QASUME-IR to a specific wavelength and keep monitoring the incoming irradiance, while we tune the laser wavelength across the spectral slit of the spectroradiometer, while nearly continuously monitoring the exact laser wavelength and irradiance using a wavemeter and reference trap detector respectively. We repeated these measurements up to 50 times per wavelength setting, in order to perform statistics on the results.

The scatter of the blue dots shown in Figure 1 at wavelengths longer than 600 nm are from the QASUME-IR measurements. We are also puzzled where it comes from, as the standard deviation of the measurements is less than the scatter between individual points. We also repeated several points on other days and with different intensity settings. Among the possible causes that could cause the observed scatter are:

a) Spectral stability of the laser line, which sometimes was unstable, and required a reset of the laser.

b) Possible instabilities of QASUME-IR and small power lamp calibration system, since the measurements shown in Figure 1) are related to each other using spectral irradiance calibrations performed typically once per day only.

The difference between the blackbody calibration and TULIP at wavelengths shorter than 400 nm seem to be systematic and were already observed in our previous campaign in 2014. We attach a figure below which shows the comparison between blackbody and TULIP in 2014 and 2022:

[Figure]

*Figure 1 Spectral irradiance ratios obtained from the blackbody and TULIP facilities, compared to the spectral irradiance obtained from a set of FEL lamps at PMOD/WRC, which were themselves calibrated by PTB relative to their blackbody in previous years. (See Gröbner J., and P. Sperfeld, "Direct traceability of the portable QASUME irradiance scale to the primary irradiance standard of the PTB," Metrologia, 42, 134—139, 2005 for more details).*

**1c)** As you correctly pointed out, the FTIR data was currently only used to assess the trace gas correction uncertainties. Future work will be needed to calibrate the FTIR data and combine it with the low-resolution spectroradiometers, as we did previously for the QASUMEFTS solar spectrum. However the FTIR data is available in the range from 950 nm to 2100 nm, and not at shorter wavelengths. The result from the Langley analyses of the FTIR data are available from the open-access project repository at https://zenodo.org/communities/19env04-mapp/

If you need additional information, we suggest to directly contact the responsible for this dataset, co-author Tom Gardiner (tom.gardiner@npl.co.uk).

**2)** I hope this was answered previously: BTS and PSR are commercial instruments which are available to interested users, and we have shown in this manuscript their potential in being able to retrieve spectral AOD, when following the procedures described here.

**3)** We are happy to rephrase this sentence, which was taken nearly literally out of the paper by Fox and Green.

**4)** This is a good point; we will add the reference to the ozone absorption cross sections which we used here. The following publication and references therein describe the impact of different ozone cross sections: Gröbner, J., Schill, H., Egli, L., and Stübi, R.: Consistency of total column ozone measurements between the Brewer and Dobson spectroradiometers of the LKO Arosa and PMOD/WRC Davos, Atmos. Meas. Tech., 14, 3319–3331, https://doi.org/10.5194/amt-14-3319-2021, 2021.

As the ozone absorption in the spectral range discussed here is relatively small (see section 5.1.2) the use of different ozone cross sections would not give a significant difference in the retrieved AOD.

**5)** This is a good suggestion, and we will add a brief outlook at the end of the manuscript.